# Conformational Properties of Flaxseed Rhamnogalacturonan-I and Correlation between Primary Structure and Conformation

**DOI:** 10.3390/polym14132667

**Published:** 2022-06-30

**Authors:** Qingbin Guo, Zhengxin Shan, Yanhui Shao, Nifei Wang, Keying Qian, H. Douglas Goff, Qi Wang, Steve W. Cui, Huihuang H. Ding

**Affiliations:** 1State Key Laboratory of Food Nutrition and Safety, College of Food Science and Engineering, Tianjin University of Science and Technology, Tianjin 300457, China; guoqingbin008322@tust.edu.cn (Q.G.); shanxzz789@163.com (Z.S.); 15635157070@163.com (Y.S.); wangnifei@tust.edu.cn (N.W.); 2Department of Food Science, University of Guelph, 50 Stone Road E., Guelph, ON N1G 2W1, Canada; kqian@alumni.uoguelph.ca (K.Q.); dgoff@uoguelph.ca (H.D.G.); steve.cui@agr.gc.ca (S.W.C.); 3Guelph Research and Development Centre, Agriculture and Agri-Food Canada, 93 Stone Road W., Guelph, ON N1G 5C9, Canada; qi.wang@agr.gc.ca

**Keywords:** dietary fibre, flaxseed, pectin, polysaccharide, RG-I, structure–function relationship

## Abstract

The pectic polysaccharides extracted from flaxseed (*Linum usitatissiumum* L.) mucilage and kernel were characterized as rhamnogalacturonan-I (RG-I). In this study, the conformational characteristics of RG-I fractions from flaxseed mucilage and kernel were investigated, using a Brookhaven multi-angle light scattering instrument (batch mode) and a high-performance size exclusion chromatography (HPSEC) system coupled with Viscotek tetra-detectors (flow mode). The *M_w_* of flaxseed mucilage RG-I (FM-R) was 285 kDa, and the structure-sensitive parameter (*ρ*) value of FM-R was calculated as 1.3, suggesting that the FM-R molecule had a star-like conformation. The *M_w_* of flaxseed kernel RG-I (FK-R) was 550 kDa, and the structure-sensitive parameter (*ρ*) values ranged from 0.90 to 1.21, suggesting a sphere to star-like conformation with relatively higher segment density. The correlation between the primary structure and conformation of RG-I was further discussed to better understand the structure–function relationship, which helps the scale-up applications of pectins in food, pharmaceutical, or cosmetic industries.

## 1. Introduction

Flaxseed (*Linum usitatissiumum* L.) is a good source of dietary fibres (22–28%, *w*/*w*), in addition to bioactive alpha-linolenic acid (~23%, *w*/*w*), protein (~20%, *w*/*w*), and phenolic compounds (~1%, *w*/*w*) [1,2]. Pectins, arabinoxylans, and xyloglucans are the major dietary fibres in flaxseed. Arabinoxylans mainly exist in flaxseed mucilage, and xyloglucans exist in the kernel, while pectins are found to be widely distributed in both flaxseed mucilage and kernel. The primary structures of flaxseed dietary fibres have been elucidated in our previous studies [3,4,5,6,7]. The water-soluble mucilage extracted from flaxseed hulls by water soaking (yield of 9.7%) were fractionated into a neutral fraction gum and an acidic fraction gum using ion-exchange chromatography. The neutral fraction consisted of high molecular weight (*M_w_*) (1470 kDa) arabinoxylans whereas the acidic fraction (FM-R) was mainly composed of rhamnogalacturonans with a higher *M_w_* fraction (1510 kDa) and a lower *M_w_* fraction (341 kDa). As shown in Figure 1A, the FM-R was highly branched, and 54 mol% of rhamnosyl units in the backbone were substituted by shorter side chains (1~3 residues containing β-D-Gal*p*, α-L-Fuc*p*, or β-D-Xyl*p* units). In addition to the flaxseed hulls, 20% of dietary fibre was also found in the flaxseed kernel, which was separated into several fractions by the sequential extraction method in our previous study [5]. The fraction obtained from 1 M KOH (FK-R) also contained an RG-I fraction, from which the rhamnosyl units were substituted by much longer arabinan side chains (up to 90 arabinofuranose units), as shown in Figure 1B [4,5].

The primary structure and conformation played vital roles in the functionality and thus affected the application of pectins. However, the information on the structure–conformation correlation of RG-I is limited from previously published papers, and no study was reported on the structural/conformational comparison of flaxseed RG-I fractions (from either kernel or mucilage). The RG-I fractions from flaxseed kernel and mucilage are supposed to demonstrate different conformational properties due to the different lengths of the side chain, although they possess the same backbone structures. In this study, the conformational characteristics of FM-R and FK-R were investigated using multi-angle light scattering instruments (under either batch or flow mode). The correlation between the primary structure and conformation of flaxseed RG-I was further discussed. The in-depth understanding of the structural characteristics of pectins not only provides novel knowledge on the structural–conformational–functional relationships but also helps bridge the gaps between lab results (e.g., physicochemical, structural, and bioactivity characterization) and scale-up industrial applications.

## 2. Materials and Methods

### 2.1. Materials

Soluble flaxseed mucilage gum was extracted from flaxseed hulls, and flaxseed mucilage RG-I fraction (FM-R) was collected after ion-exchange chromatography [8]; the extraction and purification process is shown in Figure 2A. Flaxseed kernel RG-I fraction (FK-R) was obtained from sequential extractions as described in previous studies [2,5] and shown in Figure 2B. FM-R contained 38.7 mol% galacturonic acid and the relative neutral monosaccharide composition was mainly rhamnose (38.3 mol%), galactose (35.2 mol%) and fucose (14.7 mol%). FK-R contained 14.6 mol% galacturonic acid and the relative neutral monosaccharide composition was mainly rhamnose (2.7 mol%), arabinose (64.2 mol%), galactose (11.9 mol%) and xylose (17.8 mol%). Flaxseed hulls (cultivar: Bethune) and kernels (~70% purity; cultivar: CDC Sorrel) were supplied by Natunola Health Inc. (Winchester, ON, Canada), and flaxseed kernels (>99.9% purity) were collected through further sieving and selection. The chemicals used were of reagent grade unless otherwise specified.

### 2.2. High-Performance Size Exclusion Chromatography (HPSEC)

The elution profiles of flaxseed RG-I fractions were obtained using an HPSEC system (Shimadzu Scientific Instruments Inc., MD, USA) coupled with tetra detectors, including a UV detector (Viscotek 2600), and triple detectors (incl. a refractive index detector, a differential pressure viscometer, and the low-angle (7°) and right-angle (90°) light scattering detectors) (Viscotek TDA 305, Malvern, MA, USA). The columns had two AquaGel PAA-M columns and a PolyAnalytik PAA-203 column (Polyanalytik Canada, London, ON, Canada) in series. Elution was performed at a flow rate of 0.6 mL/min using 0.1 M NaNO_3_ with 0.05% (*w*/*w*) NaN_3_, and the column/detector systems were maintained at 40 °C. Pullulan standards with a molecular weight ranging from 50 to 800 kDa (JM Science Inc., New York, NY, USA) were used to calibrate the detectors. Data were analyzed by OmniSEC software (ver. 4.6.1, Malvern, MA, USA), and some conformational parameters (e.g., weight average molecular weight (*M_w_*) and radius of gyration (*R_g_*) could be extracted.

### 2.3. Conformational Characterization of Flaxseed RG-I

#### 2.3.1. Solution Preparation

As polysaccharide molecules could be easily aggregated in water, solvent selection and dust-free solutions are critical for conformational characterization. In order to eliminate molecular aggregates, different aqueous solutions (e.g., NaCl, NaNO_3_, NH_4_OH, NaOH, or KOH at various concentrations) with increasing ion strength were tested. Dust-free solutions were prepared by consecutive filtration RG-I solutions at least four times through 0.45 μm syringe filters. The total sugar contents were tracked before and after filtration, which were determined by the Phenol-H_2_SO_4_ method [9].

#### 2.3.2. Intrinsic Viscosity Measurement

The intrinsic viscosities of flaxseed RG-I in different solutions were measured using Ubbelohde viscometer (No. 75, Cannon Institution Company, State College, PA, USA).
(1)$$\eta_{rel}=\frac{\eta }{\eta_{0}}=\frac{t}{t_{0}}$$
(2)$$\eta_{sp}=\eta_{rel}-1$$
where *η_rel_* is relative viscosity; *η* and *η*_0_ is zero-shear viscosities of the solution and the solvent, respectively; *t* and *t*_0_ is the time required for the solution and the solvent to pass through the Ubbelhode viscometer, respectively; *η_sp_* is specific viscosity.
(3)$$[\eta ]=\underset{c\to 0}{\lim}\frac{\eta_{sp}}{c}$$
(4)$$\frac{\eta_{sp}}{c}=[\eta ]+k^{\prime }{\left[\eta \right]}^{2}c$$
(5)$$\frac{\left(\ln\eta_{rel}\right)}{c}=[\eta ]-k^{\prime \prime }{\left[\eta \right]}^{2}c$$
where [*η*] is intrinsic viscosity, calculated using the Huggins (Equation (4)) and Kraemer (Equation (5)) equations; *k*′ is the Huggins constant; and *k*″ is the Kraemer constant [10,11]. The tests were conducted at 25 °C in triplicate.

#### 2.3.3. Light Scattering Analysis

Static light scattering (SLS) and dynamic light scattering (DLS) analyses followed that of previous work [6]. Briefly, SLS and DLS measurements were conducted using a laser light scattering instrument (BI-200SM, Brookhaven Instruments, New York, NY, USA) with a He-Ne laser (637 nm), a photomultiplier, a precision goniometer, and a 128-channel digital autocorrelator (BI-9000AT, Brookhaven Instruments, New York, NY, USA). SLS was carried out at angular ranges of 30°–150°, and *M_w_* and *R_g_* were determined by the Zimm plot method [12].
(6)$$\frac{K^{}c}{R_{\theta }}=\frac{1}{M_{w}}\left(1+\frac{q^{2}R_{g}{}^{2}}{3}\right)+2A_{2}c$$
where *K* is the optical contrast factor (calculated based on Equation (7)); *c* is the polymer concentration; *R_θ_* is the Rayleigh ratio; *M_w_* is weight average molecular weight; *R_g_* is the radius of gyration; *q* is the scattering vector for vertically polarized light (calculated based on Equation (8)); *A*_2_ is the second virial coefficient.
(7)$$K=\frac{4\pi^{2}n_{0}{}^{2}{(dn/dc)}^{2}}{N_{0}\lambda_{0}4}$$
(8)$$q=\frac{4\pi n_{0}\sin(\theta /2)}{\lambda_{0}}$$
where *n*_0_ is the refractive index of the solvent; *dn*/*dc* is the refractive index increment of the solution; *N*_0_ is Avogadro’s number, *λ_0_* is the wavelength in vacuum; *θ* is the scattering angle. The *dn*/*dc* values of flaxseed RG-I in the tested solutions were measured by a differential refractometer (BI-DNDC, Brookhaven Instruments, New York, NY, USA). Instrument alignment was conducted using dust-free toluene (Rayleigh ratio: 1.40 × 10^−5^ cm^−1^) as a reference before each test.

The hydrodynamic radius (*R_h_*) of flaxseed RG-I was determined by DLS at 90°, which was calculated by the constrained regularization (CONTIN) method using Brookhaven BIC Light Scattering Software. All measurements were conducted immediately after filtration, and relatively lower concentrations (i.e., 0.1~0.2 mg/mL) of flaxseed RG-I solutions were selected in DLS measurements. The structure-sensitive parameter ρ was calculated based on Equation (9) [13].
(9)$$\rho =\frac{R_{g}}{R_{h}}$$
where *ρ* is a structure-sensitive parameter; *R_g_* is the radius of gyration; and *R_h_* is hydrodynamic radius. All tests were measured in duplicate at 23 °C.

## 3. Results

### 3.1. Elimination of Flaxseed RG-I Molecular Aggregation in the Solutions

Flaxseed RG-I molecular distributions in various solutions were measured by dynamic light scattering (DLS) (Figure 3). Polysaccharides could easily aggregate in water (Appendix A), and various non-covalent interactions (e.g., hydrogen bonding, van der Waals force, ionic, and hydrophobic interactions) contribute to molecular aggregation. Different solutions with increasing ion strength, including 0.1 M NaNO_3_, 0.1 M NaCl, 0.5 M NaCl, 0.1 M NaOH, 6 M NH_4_OH, 0.5 M NaOH, or 0.5 M KOH aqueous solutions, were utilized to eliminate the aggregation for a precise characterization of flaxseed RG-I conformation.

As polysaccharide aggregates could be retained by filtration (through 0.45 μm), final concentrations could be changed, and thus the recovery rates of flaxseed RG-I before and after filtration were calculated. The recovery rates of FK-R after 4-time filtration in Milli-Q water, 0.1 M NaNO_3_, and 6 M NH_4_OH were 64.7 ± 5.2%, 79.8 ± 0.3%, and 84.4 ± 0.7%, respectively. The recovery rates of FM-R and FK-R in 0.5 M NaOH both were >95%.

The molecular size distributions of flaxseed mucilage RG-I (FM-R) ranged from 20~600 nm in either 0.1 M NaCl (Figure 3A) or 0.5 M NaCl solutions (data not shown), and those in 0.1 M and 0.5 M NaOH solutions ranged from approximately 20 to 120 nm (Figure 3B) and from approximately 20 to 90 nm (Figure 3C), respectively. The mean hydrodynamic radius (*R_h_*) reduced from 77 to 56 nm with an NaCl concentration increased from 0.1 M to 0.5 M; however, they were still much higher than those in 0.1 M or 0.5 M NaOH (~29 nm). Both of the 0.1 M and 0.5 M NaOH solution showed a stable mean *R_h_* after filtration, while the gradual increases in the mean *R_h_* from approximately 29 to 63 nm were detected within 6 h after filtration at higher concentrations (>0.5 mg/mL). The latter (i.e., 0.5 M NaOH) showed a better prevention of FM-R molecular aggregates in the solution, and no obvious degradation was detected within 24 h.

The molecular size distributions of the flaxseed kernel RG-I (FK-R) ranged from approximately 30 to 200 nm in the 0.1 M NaNO_3_ solution (Figure 3D), from approximately 30 to 80 nm in the 6 M NH_4_OH solution (Figure 3E), and from approximately 20 to 90 nm in the 0.5 M NaOH solution (Figure 3F). However, the lower mean *R_h_* of FK-R in 0.5 M NaOH might be due to the mild hydrolysis of FK-R molecules, which was confirmed by the relatively higher polydispersity in the solution (Figure 3F). It is worth noting that relatively larger molecular aggregates (~5000 nm) were also observed in the 0.5 M KOH solution (Appendix A), which might be caused by potassium ion-mediated aggregation, which requires further investigation.

As FK-R was more sensitive to alkali degradation, the stability of FK-R in 6 M NH_4_OH (stored in airtight containers at 23 °C) was also evaluated based on molecular size distribution (Appendix A) and mean *R_h_* (Appendix A), which revealed that FK-R was relatively stable during 48 h storage. Moreover, the mean *R_h_* was confirmed to be independent of the light scattering angle (Appendix A) or RG-I concentration (Appendix A).

### 3.2. Static Light Scattering (SLS) Analysis

In SLS measurements, 0.5 M NaOH was selected for FM-R characterization, and two solutions (i.e., 0.5 M NaOH and 6 M NH_4_OH) were selected to characterize and compare the conformational characteristics of FK-R. As presented in Figure 4, the Zimm plot reveals the correlation between the radius of the gyration (*R_g_*), weight average molecular weight (*M_w_*), and concentration (*c*) of flaxseed RG-I solutions. The slope of angular dependence (*θ*) at *c* = 0 refers to the mean square radius of gyration (*R_g_*^2^); the initial slope of concentration dependence at *θ* = 0 refers to the second virial coefficient multiplying by 2 (2*A*_2_); and the interception (*θ* = 0 and *c* = 0) refers to reciprocal molecular weight (1/*M_w_*) [14].

Based on the Zimm pilots of FM-R (Figure 4A) and FK-R in 0.5 M NaOH (Figure 4B) or in 6 M NH_4_OH (Figure 4C), the *A*_2_ was 9.3, 2.6, and 6.4, respectively. The positive *A*_2_ value indicates that the interaction between the molecule and the solvent is more favorable than molecular interactions in the solution (i.e., aggregation). The *M_w_* of FM-R and FK-R was calculated to be 285 kDa and 550 kDa, respectively, and the mean *R_g_* of FM-R and FK-R was 38.2 nm and 31.8 nm, respectively. It should be noted that not all light scattering conditions were selected in the Zimm plots analyses of FK-R solutions, as there might be some invisible scratches of the containers (i.e., quartz cuvettes) during SLS analyses, which lowered the accuracy at certain angles; instead, more FK-R concentrations were prepared, and more solvents were compared for reliable characterization.

### 3.3. Conformational Characteristics of Flaxseed RG-I

The conformational characteristics of FM-R and FK-R from DLS, SLS, or HPSEC analyses are summarized in Table 1. The *M_w_* of FM-R was calculated to be 285 kDa by static light scattering analysis. There were two peaks in the elution profiles from HPSEC, with *M_w_* of 1510 and 341 kDa, respectively. The higher-*M_w_* fraction should be FM-R aggregates formed in 0.1 M NaNO_3_ (i.e., HPSEC eluent), which was confirmed by light scattering analyses. The structure-sensitive parameter (*ρ*) value of FM-R was 1.3, suggesting that the FM-R molecule had a star-like conformation.

The *M_w_* of FK-R was calculated to be 550 kDa from SLS, and the ρ value was 1.21 in a good solvent (i.e., 6 M NH_4_OH). FK-R had a longer branch and a higher percentage of “hairy region” than FM-R, and the conformation of FK-R was less extended than FM-R. The conformational characteristic of FK-R in 0.5 M NaOH (Table 1) also confirmed that FK-R was partially hydrolyzed. The *M_w_* (266 kDa) was reduced by 52%, and the hydrolyzed FK-R molecule formed sphere-like conformation (*ρ* = 0.72). The intrinsic viscosity ([*η*]) was determined by the Ubbelohde viscometer at 23 °C (Equations (1)–(5)). The [*η*] of FK-R in 6 M NH_4_OH was 63 mL/g, which was much lower than that of FM-R in 0.1 M NaCl (333 mL/g). The correlation between [*η*] and the conformation of flaxseed RG-I is discussed in the following sections.

The HPSEC elution profile of FK-R is presented in Appendix A. The *M_w_* of FK-R was calculated to be 596 kDa by the light scattering detectors (at 7° and 90°) coupled with HPSEC, which was comparable with the batch-mode multi-angle SLS result (i.e., 550 kDa). As FK-R had much lower uronic acid contents (14.59%, *w*/*w*) than FM-R (38.7%, *w*/*w*) [5,8], 0.1 M NaNO_3_ solution might better shield the electrostatic effects of FK-R molecules than FM-R under a relatively higher temperature and shear force in the HPSEC columns, and no aggregation was observed in the elution profile of FK-R.

HPSEC provides a fast and high-throughput option to separate the polymer fractions and evaluate the structural characteristics. However, there are still several shortcomings of the HPSEC technique: (1) there is a limited option of eluent, which is mainly dependent on the column packing material and detectors; (2) the polydispersity of polymer fractions or other mixtures, especially those within close molecular weight ranges, which could largely affect the accuracy of HPSEC results; and (3) the characterization of the primary structure (e.g., chemical composition, linkage type, backbone, branch, etc.) or some correlated analyses (e.g., intrinsic viscosity, critical concentration, rheology, etc.) should be recommended, in order to better estimate the conformational characteristics and compensate potential errors from HPSEC analyses.

### 3.4. Correlation between Primary Structure and Conformation, and Structure–Function Relationship of Flaxseed RG-I

The correlations between structure-sensitive parameter (*ρ*) value and polymer conformation have been extensively summarised in previous publications [13,14,15]. In general, the *ρ* value decreases with the increasing branching density of a polymer molecule: homogeneous sphere conformation (*ρ* = 0.788); sphere to star-like (longer branched) conformation (*ρ* = 0.788~1.33); star-like (shorter branched) to random coil conformation (*ρ* = 1.33~1.73); randomly branched conformation (*ρ* = 1.73~2.00); and rigid rod conformation (*ρ* > 2.00). It should be noted that the types of solvent and polymer polydispersity could affect the correlations.

As presented in Figure 5, the *ρ* value of FM-R was 1.3, with *R_g_* 38.2 nm and *R_h_* 29.6 nm. The branches in the RG-I backbone were composed of shorter side chains, e.g., β-D-Gal*p*-(1→, α-L-Fuc*p*-(1→ or β-D-Xyl*p*-(1→ residue. FM-R had relatively higher uronic acid contents (38.7%, *w*/*w*), indicating a longer homogalacturonans (HG) region compared with FK-R. The ρ value of FK-R ranged from 0.90 to 1.21, and the branches in the FK-R backbone were composed of highly branched and much longer arabinans (i.e., 30~90 arabinofuranose units), suggesting the sphere to star-like conformation and relatively higher segment density.

The *ρ* value of FK-R was comparable to the water soluble soybean polysaccharide (SSPS), which was calculated to be 1.1 [16]. Both FK-R and SSPS were extracted from the cotyledons of oilseed, and they shared some common structural characteristics, such as RG-I backbone, lower uronic acid contents, and long neutral side chains [5,17]. The highly branched structure contributes to the sphere of star-like conformation, which shows a relatively lower viscosity compared to citrus pectin (i.e., HG) and has excellent emulsifying capacities to stabilize acidic dairy beverages or plant-based milks. The linkage patterns of arabinans (mainly hydrolyzed from RG-I) of different plant origin, such as legume (e.g., pea, cowpea, pigeon pea, mung bean), oilseeds (e.g., flaxseed, rapeseed, soybean, olive), vegetable (e.g., cabbage, sugar beet, carrot), and fruits (e.g., apple, grape, pear), have been summarised in earlier studies [5]. Other than SSPS, the similarity in the primary structure of oilseed RG-I might contribute to some common conformational characteristics, and further studies are still required.

As aforementioned, the intrinsic viscosity ([*η*]) of FK-R in 6 M NH_4_OH or 0.1 M NaNO_3_ (59~63 mL/g) was much lower than that of FM-R in 0.1 M NaCl (i.e., 333 mL/g). It confirmed that FM-R (with shorter side chains) were more extended than FK-R. The [*η*] reflects hydrodynamic volume occupied by a polymer molecule, which is inversely proportional to molecular density, and the [*η*] value mainly depends on the molecular weight, chain rigidity, and solvent [18]. As RG-I molecules carry charges, the force of electrostatic expansion becomes more dominant in water or diluted salt solutions; moreover, the potential molecular aggregation also contributes to the increase in intrinsic viscosity.

The short-chain fatty acids (SCFA) profiles of FK-R and FM-R are presented in Figure 5, which were evaluated through the in vitro fermentation of pig colonic digesta [19]. Compared with psyllium fibre (i.e., arabinoxylan), flaxseed RG-I were relatively slower fermentable dietary fibres, but they had a higher level of total SCFA production than psyllium fibres after 72 h incubation. FM-R and FK-R showed similar trends in acetic acid and propionic acid production, while the cultures grown with FM-R had a higher level of SCFA production. The results indicate the uronic acid content, substitution, degree of branch, side chains, molecular weight, and conformation of dietary fibres might impact the SCFA generation and fermentation rate. The structural–conformational–functional relationships of pectin from various sources, e.g., sugar beet, citrus pectin, flaxseed, and molecular simulation [20,21,22,23,24,25,26,27,28,29,30,31,32,33,34,35,36,37,38,39,40,41,42,43], are summarized in Table 2. For example, the conformational properties of RG-1 or HG under different extraction and processing conditions were also compared, e.g., alkali-extracted pectins displayed a three-dimensional structure and compact folded conformation while acid-extracted pectins possessed a relatively extended conformation [22]. Moreover, ultrasound degradation changed the structural and conformational characteristics of citrus pectin, which significantly influenced its functional properties [23]. The effects of the side chain on the conformational properties of RG-I in both solution and gel have been studied and longer side chains were associated with increased entanglements for pectin molecules [24]. Using the molecular modeling method, the location and degree of acetylation on the conformation of both RG-I and RG-II were investigated, from which acetyl groups at both O-2 and O-3 of galacturonic acid in the backbone of RG-I and HG were energetically favourable [30].

As non-starch polysaccharides cannot be hydrolyzed by human digestive enzymes, the influence of pH, dilution factor, and/or shear rate (in the GI tract) on the conformational characteristics in vivo still requires further studies. The high throughput characterizations (e.g., multi-detector HPSEC with improved column material, and conformational modelling by supercomputer), as well as the establishment of polysaccharide structural database, may help better predict conformational characteristics for various scenarios.

## 4. Conclusions

The molecular size distributions of flaxseed mucilage RG-I (FM-R) ranged from approximately 20 to 90 nm in 0.5 M NaOH solutions with a mean hydrodynamic radius (*R_h_*) of 29.6 nm, and the molecular size distributions of the flaxseed kernel RG-I (FK-R) ranged from approximately 30 to 80 nm in 6 M NH_4_OH solution with a mean *R_h_* of 26.3 nm. The weight average molecular weight (*M_w_*) of FM-R and FK-R was calculated to be 285 kDa and 550 kDa, respectively, and the mean radius of gyration (*R_g_*) of FM-R and FK-R was 38.2 nm and 31.8 nm, respectively. The structure-sensitive parameter (*ρ*) value of FM-R was calculated as 1.3, suggesting that FM-R molecules had star-like conformation, while FK-R molecules had a relatively higher segment density showing sphere to star-like conformation. Based on short-chain fatty acid (SCFA) profiles after in vitro fermentation, the primary structure (e.g., monosaccharide composition, uronic acid content, substitution, degree of branch, or side chains) and conformational characteristics (e.g., molecular weight or molecular shape) all impacted the SCFA generation and fermentation rate. In addition, the structural–conformational–functional relationships of pectin from various sources, e.g., sugar beet, citrus pectin, flaxseed, and molecular simulation, are also summarized. The correlations, such as the effects of the side chain (galactan chain), the location and degree of acetylation, and the extraction/processing conditions on the conformation of both RG-I and RG-II, helps better understand pectins at the molecular level, and could guide scale-up the applications of pectins from various plant sources.

## Figures and Tables

**Figure 1 polymers-14-02667-f001:**
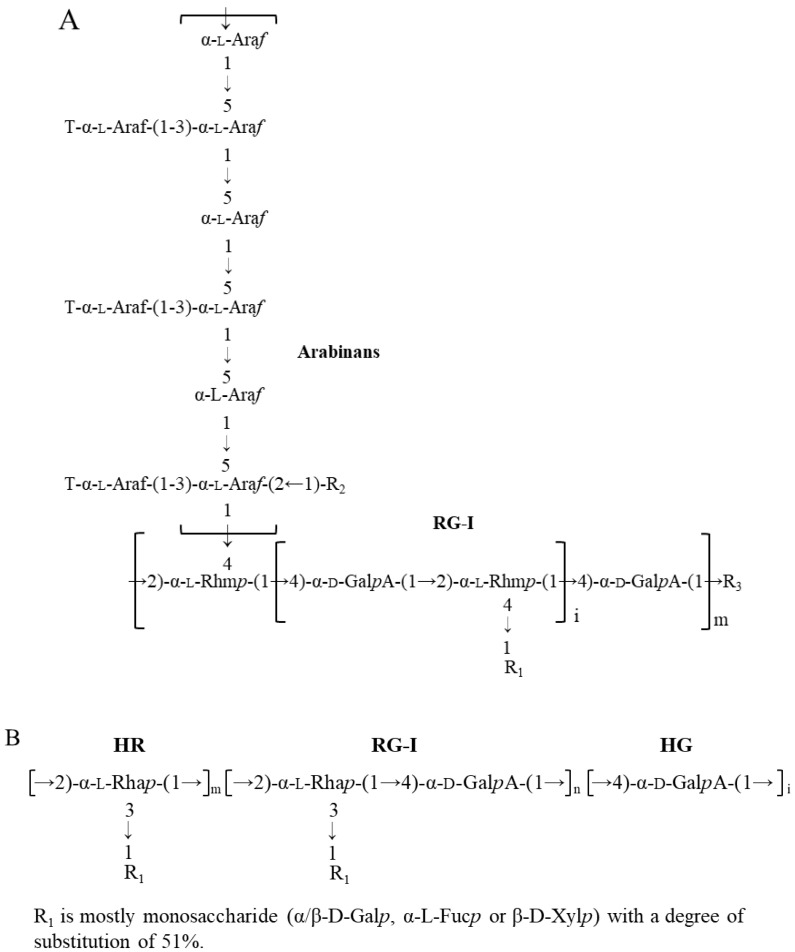
Proposed RG-I structure from flaxseed hulls ((**A**), adapted with permission from Ref. [5]) and flaxseed kernel ((**B**), adapted with permission from Ref. [4]).

**Figure 2 polymers-14-02667-f002:**
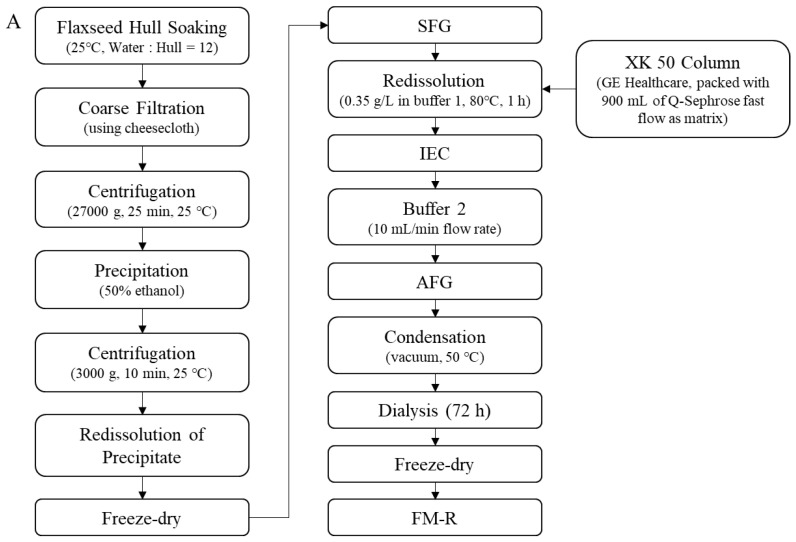
Extraction and purification process ((**A**), flaxseed hull mucilage gum, adapted with permission from Ref. [8]; (**B**), flaxseed kernel, adapted with permission from Ref. [2]).

**Figure 3 polymers-14-02667-f003:**
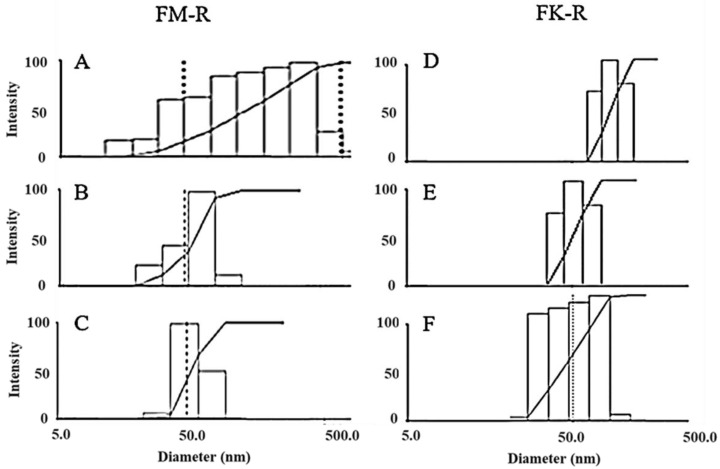
The molecular size distribution of flaxseed mucilage RG-I (FM-R) in 0.1 M NaCl (**A**), in 0.1 M NaOH (**B**), and in 0.5 M NaOH (**C**), and flaxseed kernel RG-I (FK-R) in 0.1 M NaNO_3_ (**D**), in 6 M NH_4_OH (**E**), and in 0.5 M NaOH (**F**) determined by dynamic light scattering (at 0.1 mg/mL solution concentrations, 23 °C).

**Figure 4 polymers-14-02667-f004:**
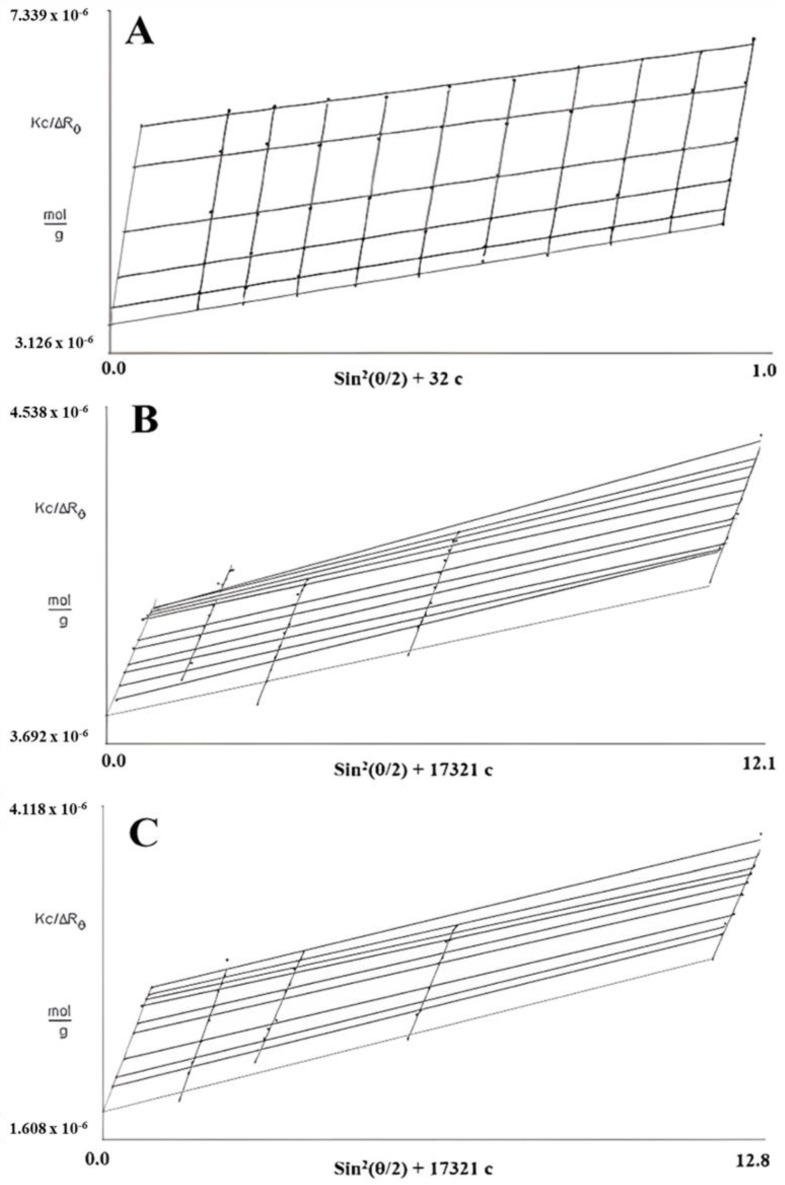
Zimm plots of flaxseed mucilage RG-I (FM-R) in 0.5 M NaOH (**A**), and flaxseed kernel RG-I (FK-R) in 0.5 M NaOH (**B**) or 6 M NH_4_OH (**C**) determined by static light scattering (at various solution concentrations, 23 °C).

**Figure 5 polymers-14-02667-f005:**
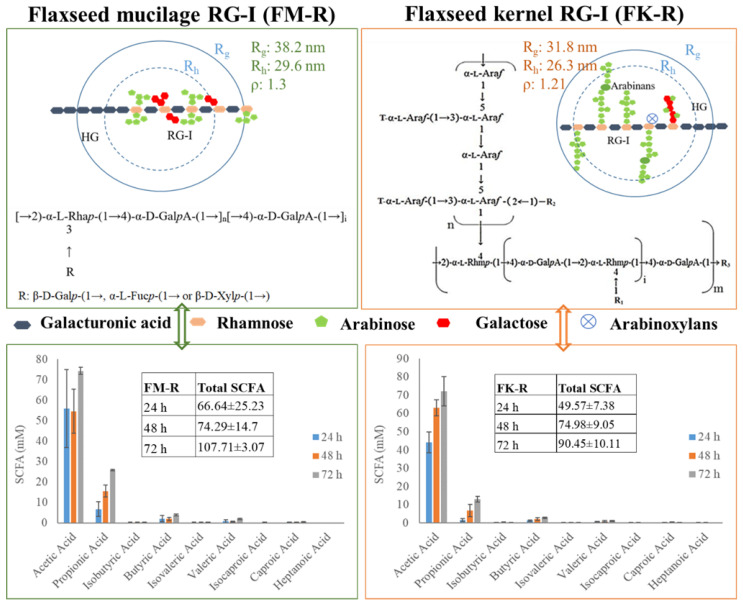
The correlation between the primary structure and conformation of flaxseed mucilage RG-I (FM-R) and flaxseed kernel RG-I (FK-R), and the correlation between the structure/conformation and SCFA profiles after in vitro fermentation; the structure of flaxseed RG-I fractions were reproduced from the results of previous studies [4,5]; SCFA profiles were reproduced from previous results [19].

**Table 1 polymers-14-02667-t001:** Conformational characteristics of flaxseed kernel RG-I (FK-RGI) and flaxseed mucilage RG-I (FM-RGI).

Conformational Characteristic/ Solution System	*M_w_* (kDa)	*R_h_* (nm)	*R_g_* (nm)	*A_2_* (10^−4^ cm^3^ mol/g^2^)	*ρ*	[*η*] (mL/g)
Flaxseed Mucilage RG-I (FM-R)						
FM-R in 0.1 M NaCl	-	-	-	-	-	333 ± 0.1 ^a^
FM-R in 0.1 M NaNO_3_ (by HPSEC)	1510 ^b^ and 341 ^b^	-	-	-	-	-
FM-R in 0.5 M NaOH	285 ± 25	29.6 ± 2.3	38.2 ± 2.7	9.3	1.30 ^c^	-
Flaxseed Kernel RG-I (FK-R)						
FK-R in Milli-Q water	1168 ± 33	88.70 ± 0.45	88.8 ± 2.4	5.00 ± 0.96	1.00 ^c^	-
FK-R in 0.1 M NaNO_3_	861 ± 7	39.10 ± 0.01	35.3 ± 1.0	0.36 ± 0.20	0.90 ^c^	59 ± 1 ^a^
FK-R in 0.1 M NaNO_3_ (by HPSEC)	596.1 ^b^	-	49.6 ^b^	-	-	-
FK-R in 6 M NH_4_OH	550.4 ± 5	26.30 ± 0.06	31.8 ± 1.5	6.36 ± 0.79	1.21 ^c^	63±1 ^a^
FK-R in 0.5 M NaOH	265.8 ± 3	24.90 ± 0.17	18.0 ± 0.6	2.55 ± 0.6	0.72 ^c^	-

^a^ data determined by Ubbelohde viscometer at 23 °C. ^b^ data determined by HPSEC coupled with tetra-detectors at 40 °C. ^c^ structure-sensitive parameter (*ρ*) value was calculated based on Equation (9).

**Table 2 polymers-14-02667-t002:** Structural–conformational–functional relationships of pectins from various plant origins.

Sources of the Pectin	Structural Characteristic	Conformation	Function	Reference
Citrus	61.80% RG-I	(1) An intertwined network; (2) Compact conformation (*R_g_*, 7.70 nm; *R_h_*, 4.78 nm).	(1) Favorable fermentation properties (high production of SCFAs, increased beneficial gut microbes, and decreased potential pathogenic bacteria); (2) A compact conformation contributed to the alleviation effects on acute colitis.	[20,21]
	(1) P3 (21.64%) and P10 (46.67%) possessed the highest RG-I content among acid- and alkali-extracted pectins, respectively; (2) P10 and P11 revealed the cross-linked networks.	(1) Acid-extracted pectins possessed a relatively extended conformation; (2) Alkali-extracted pectins displayed a three-dimensional structure and compact folded conformation.	(1) P3, which had a compact and extended conformation, exhibited the highest viscosity and ES (75%); (2) P10, which had a porous surface structure and larger (23 nm) three-dimensional nanostructure, exhibited the highest ion-binding capacity (K^+^, Ca^2+^, and Fe^3+^).	[22]
	(1) GalA: ≥ 74%, dried basis; (2) Methoxyl groups: ≥6.7%, dried basis	(1) Ultrasound degradation appreciably changed the structural and conformational characteristics of citrus pectin, which further significantly influenced its functional properties; (2) Untreated pectin had higher *M_w_*, *R_g_*, DM, and neutral sugar side chains, as well as a more extended flexible-chain conformation.	(1) Untreated pectin had more pronounced rheological properties and gel-formation capacity than ultrasound-treated pectins; (2) Ultrasonic-treated pectins had better emulsifying properties than untreated pectin; (3) Functional properties of pectin were largely determined by its *M_w_* and conformation.	[23]
	A comparison between untreated and enzymatical debranched pectin.	Longer side chains were associated with increased entanglements between pectin molecules.	Gel strength decreased for pectin gels with lower amount of side chains.	[24]
	RG-I enriched pectin contains abundant arabinan side-chains.	In acid-induced gelation, low pH promotes formation of hydrogen bonding and hydrophobic interactions within the HG region and the side-chains create a tighter conformation, eventually allowing for stronger interactions between the pectin chains.	(1) RG-I enriched pectin forms gel under both cation and acid conditions and its side-chains improve network formation; (2) Arabinan side-chains stabilize junction zones in cation-induced gel through entanglements; (3) Arabinan side-chains form dense network in acid-induced gel through hydrogen bonds.	[25]
	(1) High methoxyl pectins; (2) DE: 62.8 ± 2.02%; (3) Protein: 2.76 ± 0.26 wt%; (4) Moisture: 5.15 ± 0.23 wt%; (5) GalA: 84.80 ± 2.19 wt%; (6) Ash: 2.91 ± 0.29 wt%.	Ethanol could reduce the helix conformation and zeta potential of pectin chain, leading to compact conformation and enhanced interaction among pectin molecules.	(1) The pectin chain conformation is compressed in a binary solvent of water/ethanol; (2) The pectin in binary solvent has better emulsion stability; (3) The emulsifying properties of pectin can be improved by 21% (*v*/*v*) ethanol.	[26]
	Alkylated pectin	(1) Modified pectin has increased apparent viscosity but decreased intrinsic viscosity; (2) The decrease in intrinsic viscosity may be due to more compact conformation.	(1) The modified pectins showed better emulsifying properties (EPs); (2) EPs of modified pectin increased with the DS and alkyl chain length.	[27]
	Alkylated pectin (ALP)	(1) ALP with a higher degree of substitution (DS) had sphere conformation; (2) Original pectin (HMP) and ALP with the lowest DS had random coil conformation.	(1) Gel strength of ALP with a higher DS and longer chain length was higher than HMP; (2) Gel strength of ALP was significantly positively correlated with *M_w_*.	[28]
Various origins (molecular modeling)	Methylated pectic disaccharide 4-O-α-D-galactopyranurosyl 1-O-Me-α-D-galactopyranuronic 6,6′-dimethyl diester	(1) The iso-energy contours displayed on the ‘relaxed’ map indicate an important flexibility about the glycosidic linkage; (2) There is no significant influence of the methoxyl group on the conformational behaviour of disaccharide.	-	[29]
	Acetylation of RG-I and HG	(1) Acetyl groups at both O_2_ and O_3_ of galacturonic acid in the backbone of RG-I and HG are energetically favourable, where the most important contribution comes from an acetyl group at O_2_; (2) The presence of acetyl groups did not alter the conformational behaviour of the backbones very much.	-	[30]
	A pentasaccharide fragment of RG-I	(1) Non-reducing end of the pentasaccharide is the most flexible part of the molecule; (2) The RG backbone has a stereoregular arrangement with a fairly extended conformation.	-	[31]
	Various structural models of pectic polysaccharides	The unrefined model of the alternating co-polymer polygalacto-galacturonic acid in vacuum is consistent with the experimentally measured dimension of pectin in salt excess.	-	[32]
Flax (*Linum usitatissimum* L.) stem	(1) A complex RG-I polysaccharide with variable side chains; (2) The backbone is composed of the common GalA-Rha repeats with a high degree of branching.	A complex “secondary” structure of the polymer.	Galactanase did not change the hydrodynamic volume of flax galactan (despite considerable cleavage of Gal moieties).	[33]
Sugar beet	(1) More hydrophobic character and high protein content; (2) Interfacial structure of sugar beet pectin (SBP) studied by atomic force microscopy.	(1) SBP adsorbed at the air/water interface forms an elastic layer, as evidenced by pendant drop and surface shear rheology measurements; (2) The pectin chains prevent the formation of a densely packed protein layer.	The interfacial pectin film is more resistant to displacement by surfactants than a pure protein film, possibly because of the formation of linkages between the pectin chains.	[34]
	Acid-extracted pectin is heterogeneous with respect to molar mass, intrinsic viscosity, and composition.	(1) Fractions rich in neutral sugars have semi-flexible or random coil conformations; (2) Fractions rich in galacturonic acid have rigid rod-type conformations.	These “weight-average” molar mass, intrinsic viscosity or conformation may not necessarily be representative of the distribution of pectin molecules and this has repercussions for their functional properties.	[35]
	RG-I and HG fractions from enzymatic hydrolysis of acid extracted sugar beet pectin.	(1) RG-I had high weight average molar mass (188,000 g/mol), but low intrinsic viscosity (36 mL/g), which is consistent with a random coil conformation (L(p) = 1.4 nm); (2) HG fraction had a relatively low weight average molar mass (20,000 g/mol), but a rather high intrinsic viscosity (77 mL/g), which is consistent with the HG fraction being rigid in the solution (L(p) = 9.8 nm).	The degradation of the HG region has an important impact on intrinsic viscosity, but less on molar mass and the inverse is true for the degradation of RG-I region.	[36]
	(1) Protein: 4.1 ± 0.2%; (2) *M_w_*:104 ± 5 kDa; (3) GalA: 66.1 ± 1.8%; (4) Degree of methylation: 52.1 ± 1.7%; (5) Degree of acetylation: 23.1 ± 0.5%; (6) Trans-ferulic acid: 706 ± 21 mg/100 g.	(1) The polydispersity index of sugar beet pectin indicated a narrow distribution; (2) The conformation of SBP remains compact at pH 3 and the unfolded conformation remains compact at pH 5.	(1) A compact pectin conformation seems beneficial in stabilizing small amounts of oil; (2) In the case of high pectin-to-oil ratios, the smallest droplets were stabilized at pH 3, when SBP molecules were compact and the positive effect of fast adsorption kinetics dominated droplet stabilization.	[37]
	Low-methoxyl pectin	(1) Increasing the Ca^++^ ion concentration resulted in secondary aggregation; (2) Image after depectinization clearly showed that the main chain and branched blocks of pectin were completely split into single galacturonic acid units, whereas pectolytic enzymes remained unchanged.	Addition of Ca^++^ ions into the aqueous solution of low-methoxyl pectin caused gelation by means of salt linkages between the carboxyl groups of adjacent pectin molecules.	[38]
Flax fiber cell wall	RG-I containing the galactan side chain	(1) Extended three-fold helical structure of the RG linear backbone is the most energetically favorable motif; (2) Branching helps to stabilize a conformer of the backbone twisted along 1→2 glycosidic linkage triggering the orientation of long side chains without altering the extended overall backbone chain conformation; (3) The extended six-fold helical type of structure of the β-galactan chain displays conformational rigidity.	Neutral β-galactans lacking charged groups and displaying higher relative stiffness of helices can deeper interpenetrate and maintain the duplex structure throughout van der Waals interactions and hydrogen bonding.	[39]
	A pure RG-I without consecutive galacturonic residues and modifying groups in the backbone.	(1) Rising of the intensity of the bands attributed to galactose and glycosidic linkages in RG-I gel compared to the solution where this polymer exists as molecule associate indicates that the spatial organization of galactans in gel is changed; (2) Being destabilized at volumetric microwave heating RG-I associates are repacked, forming the network where RG-I molecules are entangled by galactan chains.	Removal of half of galactan chains from RG-I leads to loss of a gelling capability pointing out their leading role in this process.	[40]
	RG-I and its fragments, obtained after galactanase treatment	(1) Flax fiber RG-I retains hydrodynamic volume after galactanase treatment; (2) Flax fiber RG-I molecules form associates with the backbone at the periphery.	The formation of RG-I associates with the backbone at the periphery and the interaction between the side chains to form a core zone.	[41]
	Flax RG-I	(1) RG-I molecules in gel and solution possess different types of short-range structures; (2) In gel, polysaccharide chains are highly hydrated and homogeneously structured; (3) In solution, RG-I molecules self-associate with heterogeneous local packing; (4) Long-chain β-galactans form ordered tightly packed associates in solution.	(1) In colloidal solution, the side chains of RG-I are heterogeneously associated due to the constrains imposed by a stiff backbone; (2) Galactan-enriched fraction of RG-I with enzymatically cleaved backbone revealed the tendency of galactan chains to strongly associate in solution.	[42]
*Portulaca oleracea* L.	(1) The POWP-L was rich in linear HG with GalA content of 77.6%; (2) POWP-H was mainly the highly branched and acetylated RG-I pectin with relative short RG-I backbone and abundant arabinogalactan II and certain galactan side chains.	POWP-H adopted a flexible chain conformation in 0.1 M NaNO_3_ solution.	-	[43]

## Data Availability

No new data were created or analyzed in this study. Data sharing is not applicable to this article.

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
