# Peer review of "Conformational Properties of Flaxseed Rhamnogalacturonan-I and Correlation between Primary Structure and Conformation"

_polymers, 2022, doi:10.3390/polym14132667_

Round 1

Reviewer 1 Report

- Figure 1 needs to improve quality at minimal 300 dpi, because in the present forms looks very poor resolution

- 2. Materials and Methods, include information about purification materials or sentence “used as received”

- Page 4 line 111- 112. “The tests were conducted at 23 °C in triplicate”, why selected 23 °C, measurement must be at 25 °C

- Improve all Figures quality according to the international publication and Joual with high impact factor (Q1)

- Manuscript has some interesting results but doesn’t have discussion, improve discussion for all figures

- Include important results in the conclusion part

Author Response

Thanks for all valuable suggestions. Please find attached Response to Reviewers.

Reviewer 2 Report

The description of the isolation of the studied sugar systems cannot be reduced to a mere reference in the literature. It has to be described in this work even laconically, because the way of isolation is critical from the point of view of what properties the materials will have, especially when it comes to properties described by the authors. The chapter "2.1. Materials" must be expanded and optimally transformed into two "materials" with a description of commercially available used chemicals and another dedicated to isolation. It does not have to be long. This is important because, for example, I am not able to judge from the paper how the authors showed that all isolated systems are molecules with neutral charge. The authors choose to SEC eluen and standards that are relevant only to such systems and for charged molecules will not be reliable. In the same way in chapter materails is characterization of materials which in my opinion should be at the beginning of chapters results. The rest of the paper is written in a correct way and after correction of the above described aspect it can be published.

Translated with www.DeepL.com/Translator (free version)

Author Response

(The authors gave the same response as above.)
